# Constitutive Activation of RpoH and the Addition of L-arabinose Influence Antibiotic Sensitivity of PHL628 *E. coli*

**DOI:** 10.3390/antibiotics13020143

**Published:** 2024-02-01

**Authors:** Jenna K. Frizzell, Ryan L. Taylor, Lisa M. Ryno

**Affiliations:** Department of Chemistry and Biochemistry, Oberlin College, Oberlin, OH 44074, USA; jfrizzel@oberlin.edu (J.K.F.); rtaylor2@oberlin.edu (R.L.T.)

**Keywords:** antibiotic tolerance, heat shock response, minimum inhibitory concentration (MIC), PHL638 *E. coli*, L-arabinose

## Abstract

Antibiotics are used to combat the ever-present threat of infectious diseases, but bacteria are continually evolving an assortment of defenses that enable their survival against even the most potent treatments. While the demand for novel antibiotic agents is high, the discovery of a new agent is exceedingly rare. We chose to focus on understanding how different signal transduction pathways in the gram-negative bacterium *Escherichia coli* (*E. coli*) influence the sensitivity of the organism to antibiotics from three different classes: tetracycline, chloramphenicol, and levofloxacin. Using the PHL628 strain of *E. coli*, we exogenously overexpressed two transcription factors, FliA and RpoH.I54N (a constitutively active mutant), to determine their influence on the minimum inhibitory concentration (MIC) and minimum duration of killing (MDK) concentration for each of the studied antibiotics. We hypothesized that activating these pathways, which upregulate genes that respond to specific stressors, could mitigate bacterial response to antibiotic treatment. We also compared the exogenous overexpression of the constitutively active RpoH mutant to thermal heat shock that has feedback loops maintained. While FliA overexpression had no impact on MIC or antibiotic tolerance, RpoH.I54N overexpression reduced the MIC for tetracycline and chloramphenicol but had no independent impact on antibiotic tolerance. Thermal heat shock alone also did not affect MIC or antibiotic tolerance. L-arabinose, the small molecule used to induce expression in our system, unexpectedly independently increased the MICs for tetracycline (>2-fold) and levofloxacin (3-fold). Additionally, the combination of thermal heat shock and arabinose provided a synergistic, 5-fold increase in MIC for chloramphenicol. Arabinose increased the tolerance, as assessed by MDK_99_, for chloramphenicol (2-fold) and levofloxacin (4-fold). These experiments highlight the potential of the RpoH pathway to modulate antibiotic sensitivity and the emerging implication of arabinose in enhanced MIC and antibiotic tolerance.

## 1. Introduction

Quickly and quantitatively establishing the efficacy of antibiotics toward emerging strains of bacteria is paramount in the clinical and industrial sectors. Antibiotic tolerance, the temporary and reversible ability of bacteria to survive in concentrations of antibiotics that would otherwise inhibit their growth, is implicated in the failure of antibiotics to treat infections, leading to re-infection, the development and persistence of biofilms, and the emergence of antibiotic-resistant strains of bacteria [1,2,3]. When framed from a higher organisms’ health perspective, bacteria’s antibiotic tolerance is costly and deadly [4,5]. The reduced efficacy of antibiotics has led to the culling of livestock, compromised crops, and increased human deaths [6]. Bacteria that are tolerant to an antibiotic have no alterations to their genome, and higher antibiotic concentration can still kill the population; transient gene expression and metabolic changes lead to the survival and continued growth of antibiotic-tolerant bacteria and the development of persister communities: bacteria that are genetically identical to susceptible bacteria, but which can tolerate antibiotics for extended periods of time [7]. We can measure bacteria’s susceptibility to different antibiotic classes through the rapid method of the Kirby–Bauer disk diffusion assay or use more quantitatively comparative analyses of the minimum inhibitory concentration (MIC) required to eliminate 90% of the bacteria. Moreover, we can quantify the tolerance a strain exhibits after acute, brief, high-concentration exposure to antibiotics via the minimum duration for killing (MDK) assay, which is measured as a time course of growth at a concentration of antibiotic that is usually higher than the MIC [8,9,10]. In comparison, antibiotic-resistant bacteria can withstand and proliferate at high antibiotic concentrations due to an acquired genetic advantage (e.g., a mutation or collection of mutations) that proves resilience against an antibiotic [11]. This advantage is passed onto future generations, increasing a population’s resistance to antibiotic treatment. Microbiologists consider antibiotic tolerance an effective stepping stone in developing antibiotic resistance: subpopulations of tolerant bacteria can acquire genetic mutations that confer resistance [3,10,11,12,13,14,15].

*Escherichia coli* (*E. coli*) is a model organism for understanding how bacteria respond to and survive in demanding environments. *E. coli* possess a range of well-characterized stress response mechanisms that enable adaptation and survival in adverse conditions [16,17,18]. These stress responses are vital for *E. coli*’s ability to tolerate various insults, including changes in temperature, nutrient availability, pH, osmolarity, oxidative stress, and exposure to antibiotics [19,20,21,22]. Stress responses are coordinated by one or more transcription factors that change gene expression to promote survival. As an example, the heat shock response is regulated by the sigma factor σ^32^ (RpoH) when *E. coli* are exposed to high temperatures or experience other forms of damage to their proteome (e.g., unusual protein degradation, misfolding, or aggregation) [23,24,25]. Activation of RpoH begins with a high-temperature-induced configurational change at the 5′ end of *rpoH* mRNA that disrupts its secondary structure and prompts its translation to an active transcription factor. RpoH recruits RNA polymerase to the promoter regions of chaperones and degradation machinery that, upon their transcription and translation, work to restore cellular homeostasis. RpoH regulates more than 90 genes in *E. coli* [26]. RpoH associates with the protein folding chaperones DnaK and DnaJ (Hsp70 and Hsp40, respectively) and the GroEL/ES chaperonin complex, which is also a part of the RpoH regulon and upregulated with heat shock. These proteins act as cellular thermometers and regulate the heat shock response by binding to RpoH and inactivating it as a transcription factor [27]. Increased temperatures induce protein unfolding, titrating these chaperones away from RpoH and instead toward misfolded and unfolded proteins, permitting a conformational change in RpoH that discourages DnaK and other chaperone binding, freeing RpoH to bind directly to DNA. Once the cell returns to its original state and the heat shock response ceases, proteases rid the cell of conformationally changed RpoH [28].

Another example of a transcriptional response pathway that is important to cell viability in stress conditions of nutrient deprivation is coordinated by the transcription factor FliA (σ^28^, RpoF). FliA plays a role in the late-stage development of flagella, the whip-like appendages of *E. coli* that contribute to cellular motility, and chemotaxis, the movement of bacteria along a gradient of nutrients or another chemical stimulus [29]. Additionally, FliA expression also induces the formation of Type I pili; these extracellular structures aid in the adherence of bacteria to a surface during the early stages of biofilm formation [30]. Transcription of FliA is activated by the master regulator of flagellar gene expression: the FlhDC transcription complex, which is responsible for the activation and regulation of the full assembly of flagella via an incredibly complex signal transduction and assembly pathway [31]. FliA is an interesting transcription factor of study as it is involved only in the final leg of this pathway. Once transcribed and translated, FliA associates tightly with the anti-sigma factor FlgM, preventing FliA’s interaction with RNA polymerase and inhibiting its activity as a transcription factor [32,33,34]. Upon activation, FliA also has a complex but demonstrated role in biofilm formation, as our lab [35] and others [36,37,38] have explored. In *Pseudomonas putida*, downstream targets of FliA, like the chemotaxis TarR-TarS two-component system and the second messenger cyclic-di-GMP, have been implicated as directly regulating antibiotic susceptibility to several antibiotics [39]. Most recently, FliA was identified among seven other protein “hubs” in the antibiotic resistance mechanism network of the gram-negative *Stenotrophomonas maltophilia*, which has an unusually high resistance for many antibiotics and heavy metals [40].

Understanding the interplay between stress-responsive signaling and antibiotic sensitivity is essential to unravel the complexities of antibiotic action and develop strategies to overcome antibiotic tolerance and resistance using currently approved antibiotics. Here, we examine the influence of overexpressing the transcription factors RpoH and FliA via an arabinose-inducible system [35,41] on antibiotic sensitivity to three different classes of antibiotics: tetracyclines, amphenicols, and fluoroquinolones. By overexpressing transcription factors, we aim to change the protein homeostasis network and examine the effect of a global response, rather than the modulation of a single, mechanistically isolated protein, on sensitivity to these antibiotic classes [42].

We use the PHL628 strain of *E. coli*, which our lab has previously used with transcription factor overexpression to monitor biofilm formation [35,43]. PHL628 has a single point mutation in the *ompR* gene, differentiating it from its parent strain, K-12 MG1655, a standard, safe, wild-type strain of *E. coli* that has been studied for decades [44,45]. The transcription factor OmpR regulates the expression of CsgD, another transcription factor that regulates the production of CsgA. CsgA is the primary curli-specific gene that encodes curlin and, in combination with the nucleator protein CsgB, causes an overproduction of curli proteins [46,47,48]. Importantly, the transcriptome of the PHL628 strain has been sequenced and can provide insight into any potential differences we might observe in our reported MICs compared to other laboratories’ work on K-12 MG1655 [49,50]. Additionally, PHL628 provides insight into how a biofilm-prone strain of *E. coli* that is otherwise non-pathogenic is influenced by the combination of transcription factor overexpression and antibiotic treatment. We report the differing impact of classical temperature-induced overexpression of RpoH and constitutively active RpoH overexpression via the I54N mutation [51] and FliA overexpression on MIC and antibiotic tolerance.

## 2. Results

### 2.1. Disk Diffusion Assays

The influence of transcription factor overexpression in the presence of these antibiotics can be quickly and semi-quantitatively measured using classical disk diffusion assays, which correlate the size of a zone of inhibited growth with the sensitivity of a bacterial population to an antibiotic [52]. We prepared LB–agar growth plates with and without 0.1% *w*/*w* arabinose and used pre-prepared antibiotic-impregnated wafers to treat lawns of bacteria overnight (approximately 16–18 h) at 37 °C, the usual growth conditions of the assay. We compared the overexpression of a constitutively active RpoH transcription factor (RpoH.I54N) with environmental heat shock, which consisted of pre-treating the lawn of plated PHL628 cells before antibiotic wafer exposure by incubating at 47.5 °C for four hours. We measured the zone of inhibition for each of the three tested antibiotics for five experimental conditions with and without arabinose: (1) pBAD.empty-, (2) pBAD.*fliA*-, or (3) pBAD.*rpoH.I54N*-expressing PHL628, (4) thermal heat shock, or (5) control PHL628 cells (Figure 1, Appendix A). We note that, for all conditions, our bacteria are well within the “susceptible” breakpoint that is defined by the Clinical and Laboratory Standards Institute (CLSI) in their most recent guidelines for each of the three antibiotics for *Enterobacterales* (tetracycline ≥ 1.5 cm, chloramphenicol ≥ 1.8 cm, levofloxacin ≥ 2.1 cm) [53].

We observed significant differences in the zone of inhibition for each of the antibiotics under different experimental conditions. Tetracycline-treated cells had a similar zone of inhibition diameter for all conditions, except for the pBAD.*rpoH.I54N* cells in the presence of arabinose, which had a significant (33.0% ± 0.1%) increase in the inhibition diameter compared to the −arabinose control and all other conditions (Figure 1A–C). Chloramphenicol-treated cells that contained a pBAD plasmid (of any type) in the absence of arabinose all had a significantly larger zone of inhibition compared to the PHL628 control (Figure 1D, Appendix A). Additionally, for chloramphenicol-treated cells, we observed a significant decrease in the zone of inhibition for heat-shocked cells with the addition of arabinose (−Ara: 2.5 ± 0.1 cm, +Ara: 2.2 ± 0.1 cm). We also note that the chloramphenicol-treated cells overexpressing the RpoH.I54N transcription factor in the presence of arabinose all had larger average diameters for their zones of inhibition; however, due to the variability of these measurements, there was no significant difference compared to the −arabinose control. Our measurements of the fluoroquinolone levofloxacin showed a substantial decrease in the zone of inhibition with the addition of arabinose for all conditions except those cells with the pBAD.*rpoH.I54N* construct (Figure 1E, Appendix A). We noted a synergistic effect with cells that were both heat-shocked and in the presence of arabinose when exposed to levofloxacin (−Ara: 3.4 ± 0.2 cm, +Ara: 2.6 ± 0.1 cm). The zone of inhibition diameter for the heat-shocked and arabinose-treated cells was approximately 15% smaller than the arabinose-treated PHL628, pBAD.*empty*, and pBAD.*fliA* cells. pBAD.*rpoH.I54N* cells provided an exception to arabinose treatment with levofloxacin exposure by showing a significant increase in diameter, reflecting a heightened susceptibility to the antibiotic compared to other +arabinose controls.

### 2.2. Minimum Inhibitory Concentration (MIC) Determination

The results of our disk diffusion assays led us to wonder whether the minimum inhibitory concentration (MIC), the lowest concentration of an antibiotic agent required to inhibit the visible growth of a microorganism, would also differ among our experimental conditions and antibiotic classes used. We identified the MIC for our three antibiotics of interest for our study’s different vectors and control conditions at 37 °C (Table 1 and Table 2). For quantitative purposes, we measured the OD_600_ of liquid cultures incubated with varying concentrations of antibiotics after overnight (15–18 h) static incubation. We determined the MIC as the concentration of antibiotic where 10% or less growth at OD_600_ compared to the untreated control, which we confirmed also appeared as no visible growth by eye. For tetracycline, we observed that overexpression RpoH.I54N via arabinose induction reduced the MIC from 1.0 µg/mL to 0.75 µg/mL, while there was no change with arabinose addition to the empty vector control or the FliA-overexpressing cells (Table 1, Appendix A). Unexpectedly, we observed our heat shock experimental conditions with tetracycline treatment demonstrated an increased MIC from 0.75 µg/mL to 1.75 µg/mL for both control and heat shock with arabinose addition conditions (Table 2, Appendix A).

For chloramphenicol, we observed no change in MIC with the addition of arabinose for the empty vector control or with FliA overexpression and a decreased MIC (from 7.0 µg/mL to 4.0 µg/mL) with overexpression of the constitutively active RpoH.I54N transcription factor (Table 1, Appendix A). When we examined heat shock conditions, we noted a significant, five-fold increase in the MIC when arabinose was present (Table 2, Appendix A) and no change in MIC with arabinose in the absence of heat shock.

All levofloxacin-treated samples demonstrated increased MIC to differing degrees with the addition of arabinose (Table 1 and Table 2, Appendix A). FliA-overexpressing cells treated with arabinose doubled their MIC, while all other conditions experienced a three-fold increase in MIC with the addition of arabinose. This global arabinose-dependent increase in MIC correlates with the disk diffusion data for levofloxacin.

### 2.3. Minimum Duration of Killing (MDK) Assays

We were also interested in whether these transcription factors would influence the tolerance of the bacteria to high concentrations of antibiotics. To understand the influence of acute temporary antibiotic exposure in combination with our overexpression constructs, we minimally modified a traditional minimum duration of killing (MDK) assay. Specifically, we employed an MDK_99_ assay that reports on the minimum duration of killing for 99% of a population, with our overexpression constructs, heat-shocked cells, and controls to quantify their influence, if any, on the tolerances of our three studied antibiotics (Figure 2 and Figure 3, 1% threshold indicated by the dashed red line) [8,9]. We constrained the assay to three hours or less of antibiotic exposure comparison and used OD_600_ measurements to quantify growth; we report here the antibiotic concentrations that achieved MDK99 within this threshold (Table 3 and Table 4). For samples reaching MDK_99_, we spot-confirmed the results by plating the recovered culture and found no growth or growth of a singular colony on agar plates after overnight growth at 37 °C was permitted. 

The time–kill curves we generated for tetracycline (3.5 µg/mL) showed the addition of arabinose to the empty vector and pBAD.*fliA* cells did not influence the exposure time required to reduce the population below 1% (empty vector: >1 h, FliA: >2 h) (Figure 2A,B). Adding arabinose to the RpoH.I54N constitutively active construct reduced the tolerance of the cells to below 15 min, the shortest treatment period tested. In contrast, the −arabinose control reached the 1% threshold after 2 h of treatment (Figure 2C). For our heat shock and control conditions, we were required to increase the concentration of tetracycline to 6 µg/mL when arabinose was also added to reach MDK_99_ by 3 h treatment (Figure 3A,B).

Tolerance assays with chloramphenicol proved to be more complex, as the influence of arabinose, independent of vector construct, required a higher concentration of chloramphenicol (25 µg/mL for +Ara samples, 20 µg/mL for −Ara samples) to be used to reach MDK_99_ successfully (Figure 2D–F, Table 3). Adding arabinose to the acute treatment of chloramphenicol, the FliA and RpoH.I54N constructs reach 1% by 3 h, while the empty vector control reaches the threshold after 15 min treatment. For the heat shock and control experiments, we observed the presence of 12.5 µg/mL chloramphenicol in the absence of arabinose achieved MDK_99_ within three hours, while concentrations of 25 µg/mL or greater with arabinose achieved a plateauing effect of less than approximately 10% survival, without ever achieving MDK_99_ (Figure 3C,D, Appendix A). Interestingly, we observe this effect immediately (i.e., we see no significant difference in growth between t = 15 min and t = 3 h, as evaluated by an unpaired, two-tailed Student’s *t*-test).

We reproducibly observed levofloxacin reaching MDK_99_ in our experimental window with 100 ng/mL treatment concentrations for our plasmid-containing overexpression constructs (Figure 2G–I, Table 3). For our empty vector control and FliA-overexpressing cells, we observed MDK_99_ before 1 h of treatment with and without arabinose addition. For RpoH.I54N, we observed significant differences in samples with arabinose, which achieved MDK_99_ after 15 min treatment, while the non-induced (−Ara) cells achieved MDK_99_ after 1 h treatment. For our heat-shocked samples without arabinose, we reached MDK_99_ with 125 ng/mL levofloxacin after approximately one hour of treatment for both the control and the heat-shocked cells (Figure 3E,F, Table 4). 

We tested concentrations as high as 500 ng/mL levofloxacin in the presence of arabinose under these same experimental conditions and would immediately (after 15 min treatment) achieve approximately 10% survival with no notable decrease with additional incubation time (Appendix A). These tolerance experiments correlate with our observations for levofloxacin with the disk diffusion and MIC assays when in the presence of arabinose.

## 3. Discussion

We observed the constitutive activation of the RpoH pathway using the arabinose-inducible mutant, RpoH.I54N, which sensitized PHL628 *E. coli* to the antibiotics tetracycline and chloramphenicol, as evidenced by larger zones of inhibition in disk diffusion assays (Figure 1, Appendix A) and lower minimum inhibitory concentrations (Table 1 and Table 2, Appendix A). For RpoH.I54N, we also observed an increased zone of inhibition in disk diffusion assays for the antibiotic levofloxacin (Figure 1E). However, this effect appeared to be masked by the independent influence of arabinose in MIC and MDK assays. Interestingly, this effect was not observed with thermal heat shock, suggesting that the lack of negative feedback loops in RpoH.I54N signaling confer enhanced sensitivity, potentially due to mimicking chronic stress conditions. Chronic stress leads to an imbalance in cellular homeostasis pathways, such as protein synthesis, ion and other small molecule transport, and maintenance of the membrane [54,55,56,57]. Moreover, chronic stress can impose an increased energy demand on bacteria. Activation and maintenance of stress responses require energy to activate efflux pumps and synthesize cellular homeostasis components like chaperones and degradation machinery [58]. When internal energy resources become depleted, bacteria cannot sustain essential metabolic processes, ultimately leading to cell death [59].

We note the largest magnitude difference in MIC, a five-fold increase, was between the −arabinose and +arabinose heat-shocked cells treated with chloramphenicol (Table 2). This substantial increase in MIC was not observed for the PHL628 control condition with the addition of arabinose and was robustly repeated (N ≥ 3). While there have been studies looking at the influence of impaired or absent heat shock proteins (e.g., chaperones DnaK and DnaJ) with chloramphenicol treatment, there appear to be no studies examining the impact of heat shock on chloramphenicol-exposed cells [60]. Neither thermal heat shock alone nor arabinose provided the cultures with any significant deviation from the baseline MIC, and the overexpression of a constitutively active RpoH transcriptional pathway modestly *reduced* the MIC (Table 1). This suggests that adding arabinose and upregulating the heat shock pathway while maintaining endogenous feedback loops provides synergistic protection to these *E. coli* cells in the presence of chloramphenicol. Chloramphenicol inhibits protein synthesis by binding to the 50S subunit of the ribosome and inhibiting the action of peptidyl transferase, thus preventing peptide bond formation and new protein synthesis [61,62]. We induce heat shock before antibiotic treatment, allowing for the development of an enhanced protein homeostasis network ahead of antibiotic insult. Additionally, arabinose, which we discuss in greater detail below, may be inducing the expression of efflux pumps, which would enhance the efflux of chloramphenicol, among other cellular machinery promoting survival at higher concentrations of antibiotic [63,64]. We hypothesize that the enhanced protein homeostasis network induced by thermal heat shock, along with the changed metabolic and efflux profile with the addition of arabinose, compensates for the inhibition of the 50S subunit caused by chloramphenicol, promoting cell survival and growth at higher concentrations of chloramphenicol.

Our results also highlight an under-explored independent influence of the sugar L-arabinose on MIC and tolerance. While this influence is observed throughout our disk diffusion and MIC experiments, it is most apparent in our minimum duration of killing (MDK) tolerance experiments. We conducted MDK experiments over a fixed window of time (3 h) and identified a working concentration of antibiotic that would reach the minimum duration of killing for 99% of cells in culture, MDK_99,_ within that window (in most cases, specific exceptions are expanded on below). For our overexpression constructs, we observed no significant differences in the MDK_99_ for any of the three antibiotics. However, the overexpression of the constitutively active RpoH.I54N mutant results in earlier (e.g., 15 min vs. 3 h) achievement of MDK_99_ for both tetracycline and levofloxacin (Figure 2, Table 3, Appendix A). Additionally, a higher concentration of chloramphenicol (25 µg/mL vs. 20 µg/mL) was required when arabinose was present to reach the MDK_99_ threshold, though we are presently unsure why this was necessary. We observed the most striking differences in our minimum duration of killing assays when examining the heat shock experiments. Surprisingly, we did not observe these differences between heat shock and control, but rather with or without arabinose addition (Figure 3, Table 4, Appendix A). For these experiments, we required more than 2-fold higher concentration of tetracycline when arabinose was present to reach MDK_99_ within 3 h for both heat shock and control conditions. For chloramphenicol, we noted the concentrations we used were similar to those used with the overexpression vectors in the presence of arabinose, though, curiously, we were never able to achieve MDK_99_ (we were only able to achieve between 5–10% survival), which we confirmed with colony counting. For levofloxacin, we again see the dramatic influence of arabinose: even with a nearly 5-fold higher concentration of levofloxacin compared to the -arabinose controls, we see that the cells very quickly (e.g., at the 15 min time point) are at a 5–10% survival level; but, in the 3 h treatment window, they never go below this threshold. We see a similar effect with several lower concentrations of levofloxacin (Appendix A).

From these experiments, the following question emerges: how does the sugar arabinose independently influence antibiotic sensitivity, particularly to the fluoroquinolone levofloxacin? While the answer is outside this manuscript’s scope, we can look to the published literature to understand what is already known about fluoroquinolone resistance and arabinose metabolism. In clinical isolates, enhanced fluoroquinolone MICs result from mutations to the target gyrase and topoisomerase genes or increased levels of the AcrAB (part of the TolC system) and Cmr efflux pumps [65,66,67]. The enhanced MICs observed for cells treated with fluoroquinolones in combination with efflux pump overexpression were more modest compared to those found in gyrase and topoisomerase mutations. Our experimental system does not provide sufficient time for the growth of resistant mutations to accumulate [68], and each experiment is conducted from a clonal ancestral population of PHL628. Therefore, we do not think the population of cells we are studying has gyrase or topoisomerase mutations. It is more likely that adding arabinose induces changes in the expression of genes involved in its metabolism and, potentially, its efflux. Arabinose increases the expression of specific efflux pumps as a response to its catabolism and the accumulation of toxic phosphate–sugar byproducts that are a part of the pentose phosphate pathway (e.g., ribulose-5-phosphate) [69,70]. In *Salmonella enterica* serovar Choleraesuis, L-arabinose was shown to be a substrate for the TolC-dependent efflux system and could also induce the expression of that system [71]. In *E. coli*, arabinose stimulates the expression of the YdeA efflux pump but does not appear to influence the expression of the Cmr (MdfA) efflux pump, which is associated with fluoroquinolone efflux [63,64]. 

Others found combining arabinose with antibiotic treatment resulted in enhanced efficacy of the β-lactam class of antibiotics and may be related to the reduced abundance of the alarmone (p)ppGpp [72]. In this same study, no change in survival was observed with the combination of the fluoroquinolone ciprofloxacin and arabinose, though this result is likely culture-dependent. This group also examined the influence of the aminoglycoside antibiotics in combination with arabinose and found that for gentamicin, there was a two orders of magnitude increase in CFU/mL when arabinose was present independent of any other experimental conditions [73]. The influence of sugars on antibiotic resistance and efficacy has been studied for many different carbohydrate and antibiotic combinations, but the results are highly dependent upon bacterial species and strain as well as the identity of the sugar and its integration point in central metabolism [74]. Groups that have looked at the metabolome and transcriptome of sugar–antibiotic combinations, specifically with the intent of gaining insight into tolerant and persistent populations, point to some interesting commonalities: perturbance of the proton motive force (PMF), activation of central metabolism, and the importance of whether the experiment was conducted in the logarithmic growth phase versus the stationary phase [72,74,75,76]. 

Therefore, some next questions are as follows: (1) Is arabinose modulating the expression of efflux systems that specifically benefit antibiotics like fluoroquinolones? (2) Are there other antibiotic classes that use the same efflux pumps that export arabinose and could be modulated by the abundance of arabinose? (3) Is arabinose influencing the PMF or causing other perturbances in metabolism (e.g., increasing the presence of second messengers like (p)ppGpp) that are changing antibiotic susceptibility? (4) Are there other understudied sugars that might influence antibiotic susceptibility? There is clearly more to be understood between the uptake, signaling, and metabolism of arabinose and its influence on fluoroquinolone resistance and tolerance, which our lab is actively investigating [71].

## 4. Materials and Methods

### 4.1. Bacterial Growth and Strains

Ampicillin (IBI Scientific, Dubuque, IA, USA) or kanamycin (Sigma-Aldrich, St. Louis, MO, USA) antibiotics were used to select for the PHL628 *E. coli* cells (a kind gift from Anthony G. Hay, Ph.D., Cornell University) containing the desired exogenous plasmid. A 10% (*w*/*w*) arabinose stock solution was made by dissolving 1.00 g of L-(+)-Arabinose (Sigma-Aldrich) in 9.00 mL of ultrapure water and sterile filtering the solution using a 0.2 µm polyethersulfone membrane syringe filter (VWR, Radnor, PA, USA).

All *E. coli* cell cultures were grown in Luria Broth (LB), composed of 25.00 g of LB Broth Lennox (Hardy Diagnostics, Santa Maria, CA, USA) per liter of water. All the *E. coli* cell colonies were grown on LB plates composed of 25.00 g Lennox LB Broth (Hardy Diagnostics) and 12.00 g Agar Powder (Alfa Aesar, Haverhill, MA, USA) per liter of water. Depending on the experiment and type of selection needed, ampicillin, kanamycin, and/or arabinose were added to the LB media and LB plates. For the experiments involving FliA and RpoH.I54N overexpression, chemically competent PHL628 *E. coli* cells were used; chemically competent cells were prepared as previously described [77]. The pBAD18 plasmid was a kind gift from Laura Romberg, Ph.D., Oberlin College, and the pBAD.*empty* and pBAD.*fliA* constructs were prepared as previously described [35]. pBAD.*rpoH.I54N* was a kind gift from Jeffrey W. Kelly, The Scripps Research Institute [41]. 

Before any experiment, an overnight culture was made of 8 mL of LB, 8 µL of 1000× selection antibiotic (100 mg/mL ampicillin or 50 mg/mL kanamycin), and one colony of bacteria selected from the prepared agar plates. This was all combined in a 15 mL conical tube with the cap placed loosely using tape to ensure air flow and placed in a 37 °C shaking incubator for cell growth to occur overnight. For the PHL628 strain with the pBAD plasmid, ampicillin was used for selection.

### 4.2. Disk Diffusion Assay

The disk diffusion assay places an antibiotic-containing disk into the center of an agar plate inoculated with a lawn of bacteria to determine the antibiotic sensitivity in varying experimental conditions [52]. Agar plates were prepared as described by Section 4.1 with either 100 mg/mL ampicillin or 50 mg/mL kanamycin added for the pBAD plasmid strains and PHL628 strain, respectively. Additionally, plates with 0.1% (*w*/*w*) arabinose were made in the same manner, adding stock 10% (*w*/*w*) arabinose solution to reach the desired final concentration. Using a sterile inoculating loop, four–five colonies were removed from a prepared agar plate with overnight growth from glycerol stock and suspended in 2 mL of sterile saline (0.9% *w*/*v* NaCl). This solution was gently vortexed to ensure a smooth suspension and measured using absorbance (OD_600_, Eppendorf 6131 Biophotometer, Hamburg, Germany) and adjusted with saline according to a 0.5 McFarland standard to achieve an absorbance between 0.08 and 0.1. The suspension was immediately used to inoculate the plate. A sterile swab was dipped into the inoculum tube and rotated against the side of the tube above the level of solution to remove excess fluid. Then, the swab was streaked over the entire plate three times, rotating 60 degrees each time to evenly distribute the inoculum. Finally, the swab was rotated around the rim of the plate to pick up any excess liquid. The lid of the agar plate was left slightly ajar at room temperature to allow the surface of the plate to dry. For the heat shock portion of the experiment, heat shock occurred at 47.5 °C for four hours prior to disk placement; the control plate was kept at 37 °C for the same period [78]. Once this time was finished, the antibiotic disk placement resumed as described below.

Tetracycline (30 µg, Becton, Dickinson and Company, Franklin Lakes, NJ, USA), chloramphenicol (30 µg, Becton, Dickinson and Company), and levofloxacin (5 µg, Becton, Dickinson and Company) disks were used. The forceps were alcohol- and flame-sterilized. The sterile forceps were used to remove one disc from the cartridge and placed approximately two–three centimeters from the edge of the agar plate. The disc was then gently pressed to make sure it was securely attached to the agar. This was repeated a second time on the opposite side of the agar plate before replacing the lid onto the plate. Plates were inverted and incubated at 37 °C overnight (16–18 h). Biological triplicates were performed, with two technical replicates for each biological replicate. Following overnight incubation, the plates were removed and photographed in a light photo box, positioned next to a ruler for scale the next morning using either an iPhone 14 Pro or an iPhone 11 and analyzed in ImageJ v. 1.53a. The data were quantified to determine the diameter and area of the circles absent of growth around each disc from the photos. For each plate with two discs each, eight diameter and two area measurements were collected. Data were compiled in Microsoft Excel and the mean and standard deviation of the diameters were calculated. Statistics were calculated by a two-way ANOVA (analysis of variance) and post hoc Tukey tests using GraphPad Prism 10. GraphPad Prism 10 was used to graph the data.

### 4.3. Preparation of Antibiotics for MIC and Tolerance Experiments

The literature was consulted to determine an expected solubility for studied strains of *E. coli*. The protocol for antibiotic preparation involved weighing the mass of the powder antibiotic, adding the appropriate volume of solvent, ensuring the solid was appropriately dissolved, sterile filtering the solution, and performing any necessary dilutions. Solutions were prepared in 15 mL conical tubes. Sterile filtering used a 0.2 µm polyethersulfone (PES) membrane sterile syringe filter (VWR) and a sterile Plastipak 3 mL syringe. Serial dilutions were used to make lower antibiotic concentrations when necessary. Chloramphenicol (Sigma-Aldrich) was prepared at 50 mg/mL in 100% ethanol (Pharmco, Dearborn, MI, USA). The solution was vortexed before sterilization and use and was remade each day. Levofloxacin (Sigma-Aldrich) was prepared by dissolution in dimethylsulfoxide (Pharmco) at 2 mg/mL. The solution required vortexing and sonication with heat for approximately 30 min before sterilization and use, and was remade each day. Tetracycline hydrochloride (Sigma-Aldrich) was prepared in sterile water at 6 mg/mL. The solution was vortexed before sterilization and use, and was stored at −20 °C for approximately two weeks before being remade when precipitate was observed.

### 4.4. Minimum Inhibitory Concentration (MIC) Determination

Minimum inhibitory concentration (MIC) is a dilution experiment that serves to identify the lowest concentration of an antibiotic that causes cell death in a particular bacterial population. This is quantified by 10% relative growth or lower compared to a control with no antibiotic and is expressed as mg/L or µg/mL. The first steps in data collection are to establish a range of antibiotic concentrations, determined through consultation with the literature, and prepare appropriate solutions. Solutions of 3 mL were made even though only 1.6 mL was used to ensure appropriate distribution of the antibiotic and cells throughout the solution. All solutions contained live *E. coli* cells, LB, and some volume of antibiotic unless they acted as a control. Arabinose was added to half of the experimental samples to induce overexpression of transcription factors.

A starter culture was diluted 1:100 and then returned to the 37 °C shaking incubator for 2 h for further cell growth and to ensure cells were in a logarithmic growth stage. If heat shock was being assessed, then it occurred immediately after the 2 h growth at 37 °C and was an incubation at 47.5 °C for 4 h or 37 °C for 4 h for the control [78]. After 2 h (or 6 h for the heat shock experiments), the optical density at 600 nm (OD_600_) of the solutions was recorded. This OD_600_ was converted to cells per mL using the relationship OD_600_ = 1 = 8 × 10^8^ cells/mL. A final concentration of 5 × 10^5^ cells/mL of solution was used to set up the 96-well plates.

Cells were grown in clear 96-well microtiter plates (polystyrene, Corning, Corning, NY, USA) with 12 columns and 8 rows; thus, there were typically 12 different solutions made per plate with 8 replicates each. Each well contained 200 µL of solution and conical tubes were gently vortexed for several seconds before plating to make sure all contents were thoroughly mixed, and cells were dispersed. Once the plates were prepared, they were covered with a Breathe-Easy Cover (USA Scientific, Ocala, FL, USA) and placed in a 37 °C static incubator for 24 h.

Measurement of bacterial growth was determined at OD_600_ using the plate reader (Spectramax i3X or Spectramax M5, Molecular Devices, San Jose, CA, USA). The data were then saved and exported to an external drive for analysis in Microsoft Excel v. 16.80 and GraphPad Prism 10.

### 4.5. Minimum Duration of Killing (MDK) Assay

For MDK experiments, antibiotic-containing solution was prepared 10 mL at a time. This included 100 µL of 10% arabinose or sterile water, the appropriate concentration of antibiotic. However, much LB was needed to reach 10 mL. This solution was distributed in 200 µL aliquots to all the microfuge tubes used in the experiment. Cell preparation began with an overnight starter culture, which was diluted 1:100 and grown in a 37 °C shaking incubator for 2 h for further cell growth and to ensure cells were in a logarithmic growth stage. This culture was then diluted in LB to reach an OD_600_ of 0.175 or lower. We then performed two 1:100 dilutions with LB to be able to pipet approximately 100 cells per sample. The final dilution was kept on ice for the duration of the experiment to minimize cell growth. The control solution, used both for the 0 time-point as well as for overnight recovery, contained 100 µL of 10% arabinose or sterile water, 10 µL of 1000× ampicillin or kanamycin depending on which cells are used in the experiment, and 9.890 mL of LB.

The MDK protocol included the following times of exposure to antibiotics: 3 h, 2 h, 1 h, 30 min, 15 min, and 0 min. Beginning with the 3 h samples, approximately 100 cells were added to the pre-distributed 200 µL of antibiotic solution. This included duplicates for −arabinose and +arabinose for each condition. Once the cells were added, a sterile needle was used to poke a hole in the top of every microfuge tube so oxygen could enter the tubes. Cells were then incubated at 37 °C until all samples had reached the appropriate amount of exposure time. One hour into the 3 h incubation, the process was repeated for the 2 h sample and so on. When all cells finished incubation, they were placed in a microcentrifuge (VWR 2405-37 or Eppendorf 5415-D) for 10 min at 1200× *g*. To decant the supernatant, microfuge tubes were inverted and shaken five times to remove the majority of supernatant since the pellet was not visible. Finally, 200 µL of the control media was added to each of the microfuge tubes and they were all returned to the 37 °C shaking incubator overnight.

Heat shock, 47.5 °C incubation for 4 h, occurred following the 1:100 dilution and two-hour growth phase in the 37 °C incubator [78]. Hourly measurements were taken at an OD_600_ to confirm cells were continuing to grow. At the two-hour mark, a 1:50 dilution was performed to ensure continued logarithmic growth could be maintained for all four hours of heat shock. Samples were then treated identically as described above.

After incubating overnight with the control media for recovery, the microfuge tubes were removed from the incubator and the 200 µL of solution was transferred to a clear 96-well plate (Corning) to quantify. As described in Section 4.4 for MIC data quantification, an OD_600_ measurement of the plates was taken using the plate reader (Spectramax i3X or Spectramax M5, Molecular Devices) to determine bacterial growth. The data were saved onto the external hard drive and analyzed using Microsoft Excel v. 16.80. The compiled averages were used to identify the relative percentage of growth compared to the control with no antibiotic added, where we attempted to identify concentrations that had less than 1% growth by the 3 h mark and some growth at the 15 min mark. GraphPad Prism 10 was used to graph the time–kill curves and plot the averages and standard deviations.

## 5. Conclusions

Overexpression of the constitutively active transcription factor RpoH.I54N modestly reduces the MIC of the antibiotics tetracycline and chloramphenicol for PHL628 *E. coli*. In comparison, thermal heat shock has minimal effect on MIC and MDK_99_, with the important exception of chloramphenicol, where heat shock in the presence of arabinose increased the MIC 5-fold. Our most exciting observation was that arabinose appears to independently increase the MIC and MDK_99_ of all the antibiotics studied under varying conditions, though we see the effect most consistently with the fluoroquinolone levofloxacin. We observed a two-to-three-fold increase in MIC for levofloxacin and similarly increased MDK_99_ when in the presence of arabinose, irrespective of other experimental conditions, highlighting the independent influence of arabinose on antibiotic sensitivity. These experiments demonstrate a need for exploring the impact of arabinose and other sugars on antibiotic resistance and tolerance and the importance of the RpoH stress-responsive signaling pathway as a putative target for activation as a mechanism for enhancing *E. coli* sensitivity to existing antibiotics.

## Figures and Tables

**Figure 1 antibiotics-13-00143-f001:**
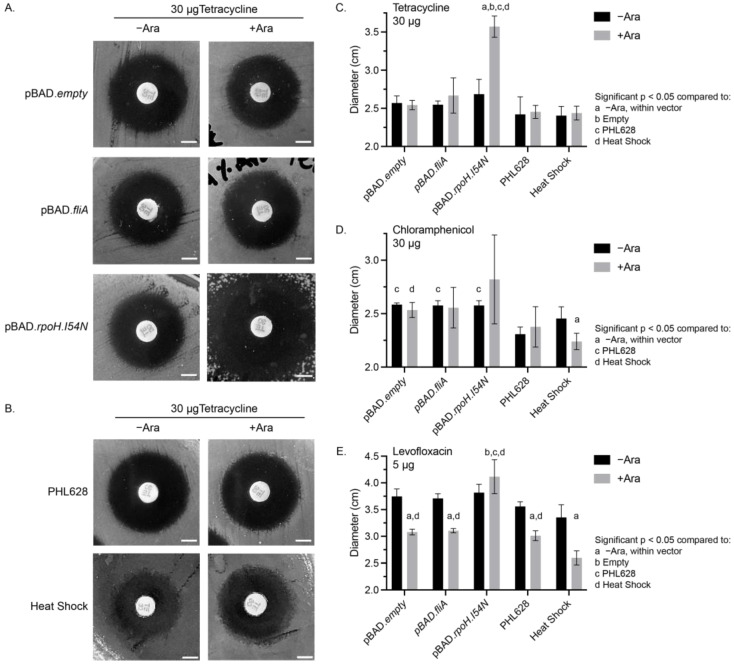
Disk diffusion assays. Representative images from disk diffusion assays for 30 µg tetracycline-impregnated disks of (**A**) plasmid-containing cell constructs, or (**B**) PHL628 control or heat- shocked cells with or without 0.1% (*w*/*w*) arabinose. White scale bar is 0.5 cm. Quantitative analysis of the average diameters of different experimental conditions for (**C**) 30 µg tetracycline-, (**D**) 30 µg chloramphenicol-, and (**E**) 5 µg levofloxacin-treated disk diffusion plates with and without 0.1% (*w*/*w*) arabinose. Significance (*p* < 0.05) between conditions was determined using a two-way ANOVA and post hoc Tukey test for N ≥ 3 samples.

**Figure 2 antibiotics-13-00143-f002:**
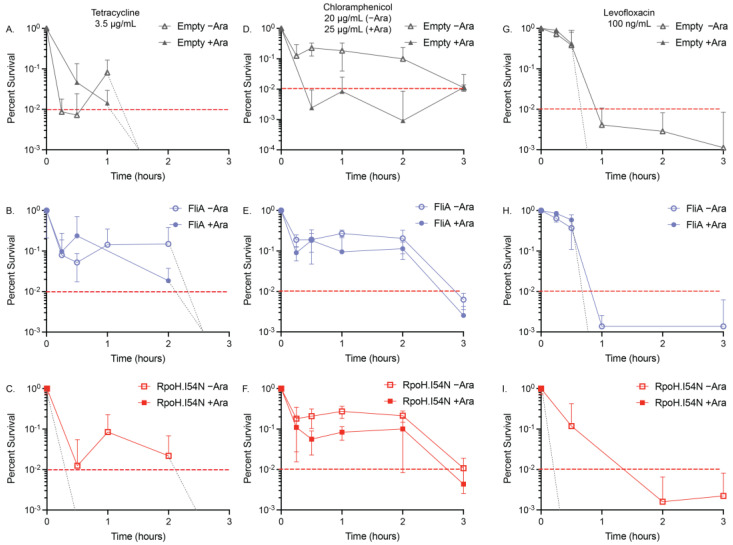
MDK_99_ time–kill curves for overexpression constructs. The acute effect on viability of our transcription factor-overexpressing constructs with (**A**–**C**) 3.5 µg/mL tetracycline +/− arabinose (Ara), (**D**–**F**) 20 µg/mL (−Ara) or 25 µg/mL (+Ara), and (**G**–**I**) 100 ng/mL levofloxacin +/− arabinose (Ara) was confirmed using relative OD_600_ measurements of N ≥ 3 biological replicates. Red dashed lines are at MDK_99_. Light grey dashed lines indicate where the time–kill curve crosses the MDK_99_ line in the absence of any survivors (see Appendix A for linear plots of the data).

**Figure 3 antibiotics-13-00143-f003:**
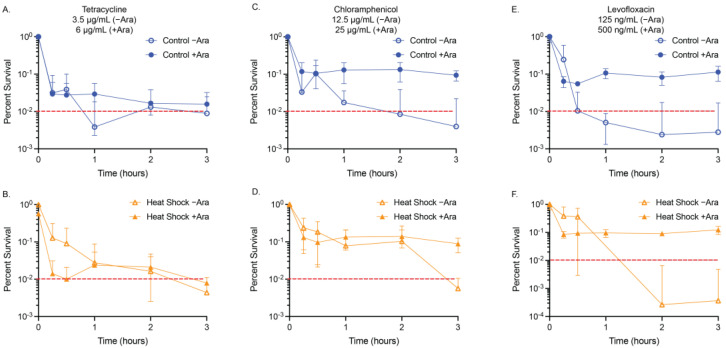
MDK_99_ time–kill curves for heat-shocked cultures. The acute effect on viability of our control and heat shocked cells with (**A**,**B**) 3.5 µg/mL (−Ara) and 6 µg/mL (+Ara) tetracycline, (**C**,**D**) 12.5 µg/mL (−Ara) or 25 µg/mL (+Ara) chloramphenicol, and (**E**,**F**) 125 ng/mL (−Ara) or 500 ng/mL (+Ara) levofloxacin was confirmed using relative OD_600_ measurements of N ≥ 3 biological replicates. Red dashed lines are at MDK_99_ (See Appendix A for linear plots of the data).

**Table 1 antibiotics-13-00143-t001:** Minimum inhibitory concentration (MIC) table for plasmid-containing PHL628 cultures with and without 0.1% (*w*/*w*) arabinose induction. Values were determined from N = 3 biological replicates.

MIC (µg/mL) at 37 °C						
	pBAD.*empty*		pBAD.*fliA*		pBAD.*rpoH.I54N*	
	−Ara	+Ara	−Ara	+Ara	−Ara	+Ara
Tetracycline	1.0	1.0	1.5	1.5	1.0	0.75
Chloramphenicol	7.0	7.0	7.0	7.0	7.0	4.0
Levofloxacin	0.05	0.15	0.05	0.10	0.05	0.15

**Table 2 antibiotics-13-00143-t002:** Minimum inhibitory concentration (MIC) table for PHL628 control and heat shocked cultures with and without 0.1% (*w*/*w*) arabinose induction. Values were determined from N = 3 biological replicates.

MIC (µg/mL) at 37 °C				
	PHL628		Heat Shock	
	−Ara	+Ara	−Ara	+Ara
Tetracycline	0.75	1.75	0.75	1.75
Chloramphenicol	7.0	7.0	5.0	25
Levofloxacin	0.05	0.15	0.05	0.15

**Table 3 antibiotics-13-00143-t003:** Minimal duration of killing (MDK_99_) table for plasmid-containing PHL628 cultures with and without 0.1% (*w*/*w*) arabinose induction. Values were determined from N = 3 biological replicates. Asterisks indicated a * 5% viability was reached.

MDK_99_ (µg/mL) at 37 °C						
	pBAD.*empty*		pBAD.*fliA*		pBAD.*rpoH.I54N*	
	−Ara	+Ara	−Ara	+Ara	−Ara	+Ara
Tetracycline	3.5	3.5	3.5	3.5	3.5	3.5
Chloramphenicol	20	25	20	25 *	20	25
Levofloxacin	0.10	0.10	0.10	0.10	0.10	0.10

**Table 4 antibiotics-13-00143-t004:** Minimal duration of killing (MDK_99_) table for PHL628 cultures with and without 0.1% (*w*/*w*) arabinose induction. Values were determined from N = 3 biological replicates. Asterisks indicate a * 5% or ** 10% viability was reached.

MDK_99_ (µg/mL) at 37 °C				
	PHL628		Heat Shock	
	−Ara	+Ara	−Ara	+Ara
Tetracycline	3.5	6.0	3.5	6.0 *
Chloramphenicol	12.5	25 **	12.5	25 **
Levofloxacin	0.125	0.50 **	0.125	0.50 **

## Data Availability

The data presented in this study are available in the article and Appendix A.

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
