# Peer review of "Constitutive Activation of RpoH and the Addition of L-arabinose Influence Antibiotic Sensitivity of PHL628 E. coli"

_antibiotics, 2024, doi:10.3390/antibiotics13020143_

Round 1

Reviewer 1 Report

Comments and Suggestions for Authors

In this article submitted by Jenna K. Frizzell, et al., and entitled "Constitutive activation of RpoH and the addition of L-arabinose influence antibiotic sensitivity of PHL628 E. coli", the author investigates the impact of overexpressing two transcription factors, FliA 13 and RpoH.I54N, in E. coli on its susceptibility to antibiotics. There are many questions that the author needs to address.

1. Line 38-39, in the section of introduction, the author state that “The reduced efficacy of antibiotics has led to the culling of livestock, compromised crops, and increased human deaths”, the reduced efficacy of antibiotics can lead to compromised crops, which is a source of confusion for me.

2. The introduction section is excessively verbose, and I think that the third paragraph (line 82-102) is unnecessary. Because the research topic of this paper is related to Escherichia coli rather than Pseudomonas aeruginosa. Moreover, the 5th paragraph (line 125-146) is also unnecessary. This is just my personal suggestion; the author can make the decision themselves.

3. Line 170, “16 – 18 hours” should replaced with “16 – 18 h”. Please review and amend the aforementioned text for any similar errors.

4. The key data associated with the overexpression of FliA and RpoH.I54N genes is missing in this article. For instance, the results of constructing the overexpression vector and screening for positive clones should be included in either the main text or supplementary materials.

5. Line 165-180, this paragraph should be placed in the discussion section.

6. Line 179, the breakpoints of the antibiotics against the specific pathogens, should be provided here.

7. Line 174 and line 263, these two sentences are only the description of results, so what is purpose of the author citing references [71] and [7-8] here?

8. Discussion section needs a revision. The results should be sufficiently discussed and supported by literature.  

9. The "Conclusions" section should be more succinct and emphasized.

Reviewer 2 Report

Comments and Suggestions for Authors

Comments:

The manuscript reported how different signal transduction pathways in the gram-negative bacterium Escherichia coli (E. coli) influence the sensitivity of the organism to antibiotics from three different classes: tetracycline, chloramphenicol, and levofloxacin. Two transcription factors FliA and RpoH.I54N were overexpressed on the PHL628 strain of E. coli to determine their influence on the minimum inhibitory concentration (MIC) and minimum duration of killing (MDK) concentration. While FliA overexpression had no impact on MIC or antibiotic tolerance, RpoH.I54N overexpression reduced the MIC for tetracycline and chloramphenicol but had no independent impact on antibiotic tolerance. For these good results, I suggested the manuscript to be published. There are too many references, it is suggested to delete some and keep around 60.

Reviewer 3 Report

Comments and Suggestions for Authors

The paper is of high quality in topic, methodology and presentation, at least at the first glance. However, I detected several major and minor discrepancies in some descriptions and in statistics:

L179/L818 - the citation [72] of the most recent guidelines is not complete and 2012 definitely not the most recent version

Figure1: the depiction of statistically significant differences is confusing. However, it is clear that all the experimental groups were compared to each other, not just the ara+ and ara- pairs. T-test is not appropriate for comparison of so many experimental groups. ANOVA or 2-way ANOVA with interactions (with Arabinose as the second factor) would be more appropriate. Different letters would serve as better indicators of the differences.

Table 2+4: both columns for PHL628 are named as ara-

L359+L361+L372 zone of inhibition, not zone of killing

L341-L429: most of the Discussion just re-narrates the results. As the authors are most focused on arabinose, I recommend to add some publications on its effect on antimicrobial resistance (doi: 10.1074/jbc.M112.420380, doi: 10.1099/mic.0.000152).

L508 - discs were placed 1 cm from the edge of the plate. This is quite close to the edge and does not correspond with the pictures of the zones and some of the largest diameters reported - a 4.5 cm zone would be only a semi-circle at best.

L559 as far as I know, neither OD=1 nor McFarland=1 are considered equal to 8 x 10^8 CFU/ml for E. coli

Reviewer 4 Report

Comments and Suggestions for Authors

Frizzell et al, presented a research article, “ Constitutive activation of RpoH and the addition of L-arabi-nose influence antibiotic sensitivity of PHL628 E. coli”. This research investigates the influence of signal transduction pathways, specifically FliA and RpoH, on antibiotic sensitivity in E. coli, revealing the potential of RpoH in modulating sensitivity and the unexpected implications of L-arabinose on antibiotic tolerance. Before acceptance, this research needs to be thoroughly revised with consideration of the following points.

1.      The abstract section, pleaseinclude numerical values for the synergistic increase in MDK99, providing a concise, impactful summary of experimental outcomes. Additionally, Specify the antibiotics affected by RpoH.I54N and arabinose, enhancing clarity and understanding for readers unfamiliar with the experiment.

2.      Introduction section can be further improved by incorporation the following research :doi.org/10.1016/j.ijbiomac.2023.127247, doi.org/10.1208/s12249-023-02620-w

3.      The manuscript mentions the surprising influence of arabinose on antibiotic sensitivity, particularly with levofloxacin. The discussion could be expanded to address the potential mechanisms through which arabinose might independently impact antibiotic susceptibility. This would enhance the reader's understanding of the observed effects.

4.      The study references previous works on antibiotic resistance mechanisms but does not explicitly connect these findings with the current results. Providing a more direct link between the observed effects and existing literature on fluoroquinolone resistance and arabinose metabolism would strengthen the discussion section.

5.      The manuscript notes a significant increase in MIC for chloramphenicol with heat shock in the presence of arabinose. Provide additional insights or hypotheses on why this specific combination leads to a substantial increase in MIC. Consider discussing potential interactions between heat shock, arabinose, and chloramphenicol in more detail.

6.      Ensure consistency in terminology and formatting throughout the manuscript.

.

Round 2

Reviewer 1 Report

Comments and Suggestions for Authors

The article submitted by Jenna K Frizzell et al., and entitled "Constitutive activation of RpoH and the addition of L-arabinose influence antibiotic sensitivity of PHL628 E. coli", was revised carefully by the authors following the review comments, the manuscript has been significantly improved. Therefore, I fully agree that this paper in its current state can be accepted.